# A Safe and Efficient Mining Method with Reasonable Stress Release and Surface Ecological Protection

Zhenghu Li [1,2], Junhui Zhang [1]🄳, Hui Chen [1], Xiuzhi Shi [1,*], Yanyang Zhang [1] and Yanjun Zhang [2]

1   School of Geology and Mining Engineering, Xinjiang University, Urumqi 830046, China;
    lzh622822@126.com (Z.L.); zhangjunhui@xju.edu.cn (J.Z.); chenhui@xju.edu.cn (H.C.); zyy@xju.edu.cn (Y.Z.)
2   Shenmu Zhangjiamao Mining Co., Ltd., Shaanxi Coal Chemical Industry Group, Xi'an 710000, China;
    ca506116342@163.com
*   Correspondence: baopo@csu.edu.cn

**Abstract:** Coal is an important basic energy source, widely distributed throughout the world, but resource abundance is uneven. Despite the need to develop and form new energy sources, coal energy maintains its dominant position. However, due to the uneven distribution and non-renewable nature of coal resources, the relationship between the supply and demand of coal resources is tight. The rational exploitation of coal and reducing resource mining wastes are particularly important at the present stage. The original mining method of the Zhangjiamao coal mine resulted in a large waste of coal resources. After replacing the "110 construction method", the original advanced end-support was canceled, which saved a lot of process time and engineering costs and greatly improved the mine production efficiency. With an average mining depth of +300 m, the working face is in a safe and stable state, and the 110-mining process has little impact on surface subsidence. Its successful application provides a reference experience for other mines to promote resource-saving and efficient mining.

**Keywords:** non-renewable; clean coal technology; waste of resources; "110 construction method"; collapse settlement control





## 1. Introduction

Coal is a solid combustible mineral. It is formed by ancient plants buried underground through complex biochemical and physicochemical changes (Figure 1). Coal is also known as black gold or industrial blood. Coal is an important basic energy source and the raw material foundation of the steel, cement, chemical industry, and other industries. In 2012, coal accounted for a record 29.9% of the global disposable energy consumption [1]. In China, coal accounted for about 70% of the disposable energy consumption structure (Table 1). According to the data of relevant departments, coal will account for more than 50% of China's energy structure by 2050 [2]. China's coal pollution is also very serious. Eighty-five percent of coal is directly used for combustion, which once made China a typical soot-polluted environment. Coal burning emits a large amount of $SO_2$, and 30% of China's land has been affected by acid rain [3]. Many studies show that in the next 10~20 years, with the development of clean coal technology (Figure 2), coal will surpass oil and become the largest energy consumption in the world. At the executive meeting of the State Council held in 2021, the Chinese government set RMB 200 billion ($3.14 billion) of special refinancing to support the clean and efficient utilization of coal based on the early establishment of financial support tools for carbon emission reduction to form a policy and promote green and low carbon developments. In recent years, with the acceleration of coal energy consumption, the mining depth of coal mines has also grown rapidly. Based on the analysis of the current situation and the existing problems of deep surrounding rock control and intelligent mining technology, Kang et al. [4] discussed key scientific problems and technical ideas. These scientific problems mainly focused on safe and efficient mining and the interaction between the roadway and stope. The technical ideas mainly focus

on reasonably increasing the length of the working face, realizing intensive production, reducing the tunneling rate, and improving the coal recovery rate [5,6].

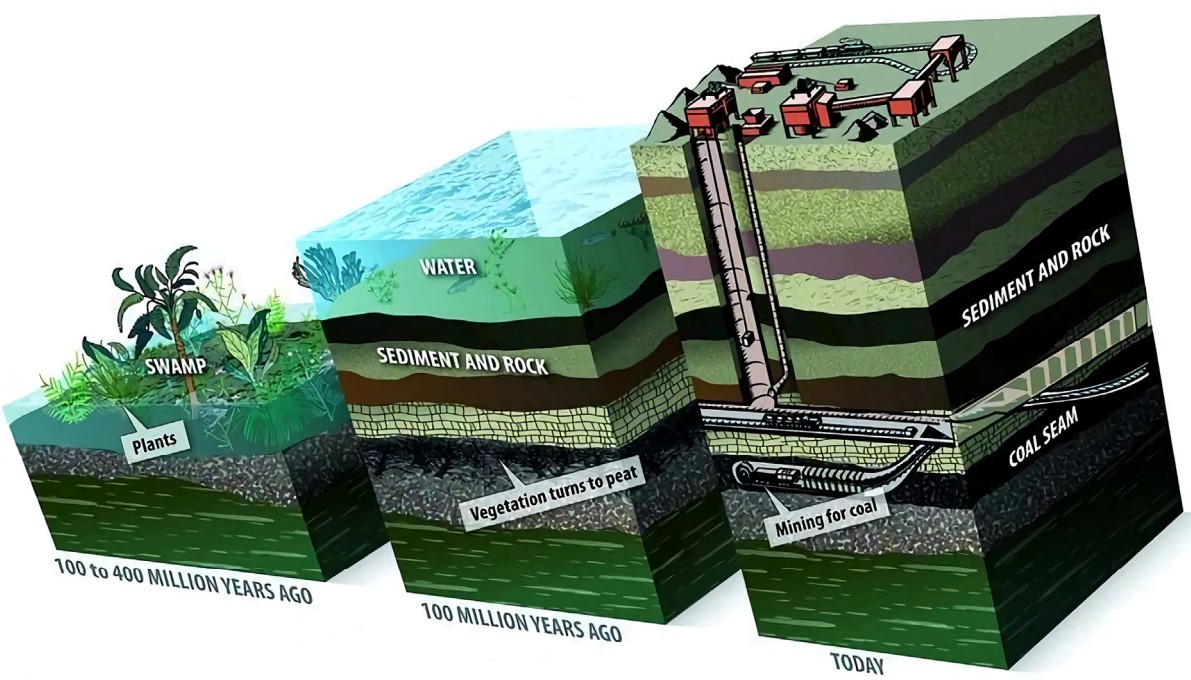

**Figure 1.** Formation process of coal resources.

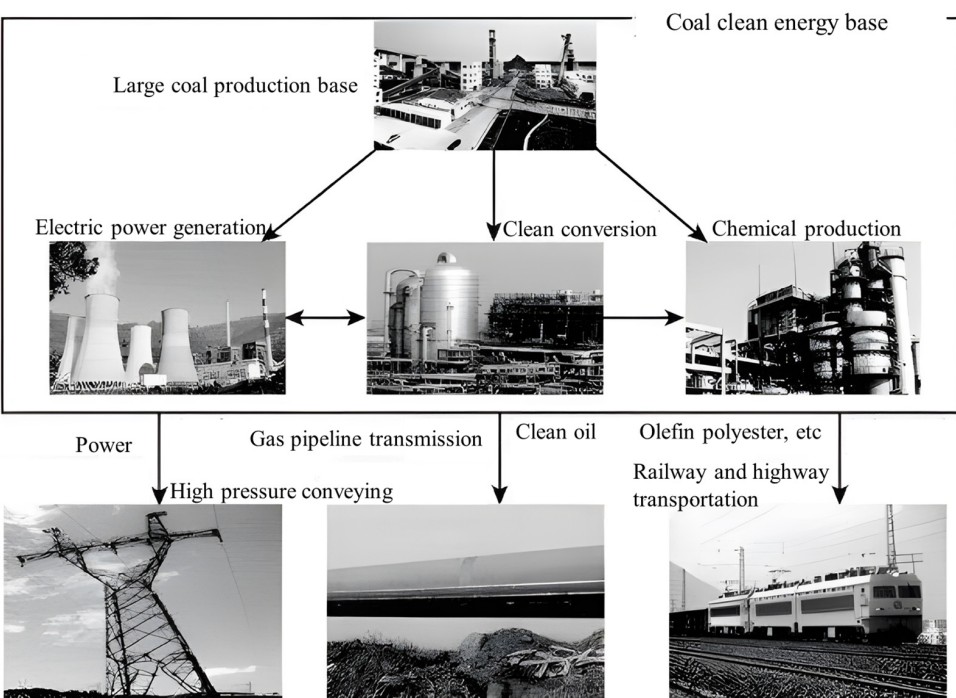

**Figure 2.** Application of clean coal technology.

The traditional utilization of coal in China is inefficient, and the power industry has become the main emission source of $CO_2$. It is urgent to build a low-carbon power industry system. At the same time, there is serious waste in China's coal mining due to technical or human factors. In the environment of promoting comprehensive energy utilization, this problem must be improved. From the evolution and development trend of coal science and technology, the integration of coal, mine construction, and underground

space development and utilization are inseparable. Some scholars have put forward a roadmap for coal technology revolution in the whole industrial chain based on four aspects: scientific mining, near-zero ecological damage, clean and low-carbon utilization, mine construction, and comprehensive utilization of underground space [7–9].

Coal resources will still occupy the status of the basic national energy source for a long time in the future. To solve the inequality between coal resource distribution areas and consumption areas, the Chinese government should vigorously develop the transportation industry and improve the transportation capacity of the central and western regions. China's major coal-producing areas have been concentrated in east and south China for a long time. Due to the distribution of railway transportation capacity, coal supply, especially power coal supply, is relatively tight in most provinces and cities in central Hunan, Hubei, Jiangxi, and some western regions. The Chinese government should adjust the structure of railway transport capacity, increase investment in railway construction, and rationally allocate the country's coal resources [10].

Because of the strategic position of coal resources, the globalization of the coal trade has become an inevitable trend. Coal resources are widely distributed all over the world. In the northern hemisphere, two huge world-class coal accumulation belts are the most prominent. One stretches across Eurasia, starting from Britain in the west, passing through Germany, Poland, and the former Soviet Union in the east, extending eastward to the east of North China and Russia's far east. The other extends in the east–west direction in north-central North America, including the coal fields of the USA and Canada. Coal resources in the southern hemisphere are mainly distributed in temperate regions, and coal resources in Australia, South Africa, Botswana, and Mozambique are affluent. The recoverable reserves of the world's major coal resource countries are shown in Figure 3. The proportion of coal resources in a country's disposable energy structure reflects the position of the coal industry in the country's energy consumption, which is also the basic factor determining the development of the country's coal industry. The composition of disposable energy structure and the proportion of coal in the world's major countries are shown in Table 1. China's coal resources account for 69.3% of the country's energy. Coal will continue to be China's primary energy source for a long time. Therefore, it is of great practical significance to reduce the waste of coal resources and realize the efficient and economic mining of coal resources. The COVID-19 pandemic had a great impact on the global energy market. The reduction in disposable energy and carbon emissions reached a new high since World War II. However, the important position of coal in traditional energy will continue for a long time.

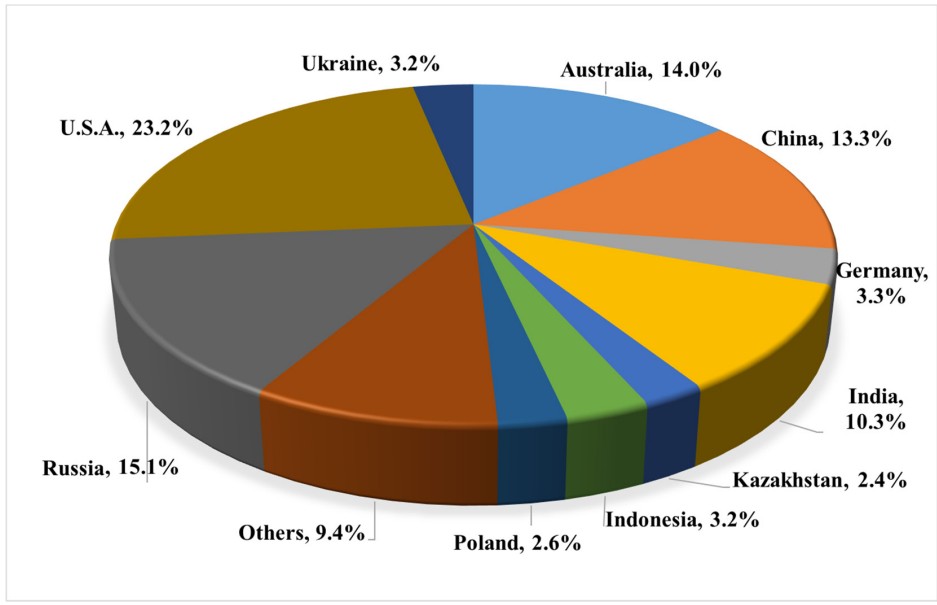

**Figure 3.** Proportion of proved recoverable reserves in major coal resource countries [11].

Table 1. Composition of disposable energy structure and proportion of coal in major countries in the world/million tons of oil equivalent [12].

| | Petroleum | Natural Gas | Coal | Nuclear Energy | Hydropower | Total | Proportion of Coal |
|---|---|---|---|---|---|---|---|
| South Africa | 26.9 | 3.4 | 89.8 | 3.2 | 0.4 | 123.7 | 72.59% |
| China | 483.7 | 129.5 | 1873.3 | 22 | 194.8 | 2703.3 | 69.30% |
| Kazakhstan | 12.8 | 8.5 | 35 | — | 1.8 | 58.1 | 60.24% |
| Poland | 25.1 | 14.9 | 54 | — | 0.5 | 94.5 | 57.14% |
| India | 171.6 | 49.1 | 289.3 | 7.5 | 6.2 | 543.7 | 53.21% |
| Australia | 46.7 | 22.9 | 49.3 | — | 4.1 | 123 | 40.08% |
| Indonesia | 71.6 | 32.2 | 50.4 | — | 2.9 | 157.1 | 32.08% |
| Republic of Korea | 108.8 | 45 | 81.8 | 34 | 0.7 | 270.3 | 30.26% |
| Germany | 111.5 | 67.7 | 79.2 | 22.5 | 4.8 | 285.7 | 27.72% |
| Japan | 218.2 | 105.1 | 124.4 | 4.1 | 18.3 | 470.1 | 26.46% |
| U.S.A. | 819.9 | 654 | 437.8 | 183.2 | 63.2 | 2158.1 | 20.29% |
| Russia | 147.5 | 374.6 | 93.3 | 40.3 | 37.8 | 693.5 | 13.45% |
| Columbia | 12.7 | 8.9 | 4 | — | 10.8 | 36.4 | 10.99% |

Data Source: BP world energy statistics yearbook, 2013.

## 2. Need for Research

China's coal resource reserve is very important and related to the development of the national economy and society. Due to the non-renewable nature of coal, the rational utilization of coal resources is very important for China to achieve sustainable development. In the process of mining coal resources, enterprises generally have the behavior of "mining fertilizer and losing thin". When the coal price rises, enterprises generally have the phenomenon of blindly expanding production. To promote coal production enterprises to save resources and reasonably develop resources, the state should establish and improve relevant systems as soon as possible to curb the waste of coal resources in all aspects. The long-standing common problems in the process of coal mining are resource waste and ecological damage. With the global energy shortage and the increasing importance of the ecological environment, the rational exploitation and utilization of coal resources have become more important than ever. Starting from the important position and role of the coal industry in China's economic and social development, Dr. Ouyang studied the contradictory relationship between the development of the coal industry and the development of the national economy and the cost of resources and environment, and proposed that the intensive development of China's coal industry under the constraints of resources and environment is feasible and beneficial [13]. Zhang comprehensively and objectively analyzed the current situation of China's coal resources, pointed out many problems existing in the process of coal resource exploration and coal mining, and put forward ideas and countermeasures to improve the guarantee degree of coal resources and rational development and utilization. These opinions are of great significance for promoting the comprehensive, coordinated, and sustainable development of China's coal industry [14]. Liao and Qian first put forward the concept of realizing the green mining of coal resources, along with scientific research a and technical framework, and then put forward the academic viewpoint of realizing scientific mining, comprehensively describing the important progress made in breaking through the concept of traditional coal mining technology [15].

## 3. Problems and Countermeasures for a Working Face

The 14,211-mining face is the 11th fully mechanized mining face of the $4^{-2}$ coal in panel 1 of the Zhangjiamao company (Northern Shaanxi mining); the average buried depth of the working face is +300 m. The working face is a double lane layout with one entry and one return. The designed advancing mining dip length is 2370.3 m, the strike length is 305.7 m, and the mining area is 724,600.7 m$^2$. The geological reserves of coal in the mining area are 6.83 million tons. The one-time inclined longwall mining method and comprehensive mechanized mining technology were adopted in the 14,211 fully mechanized mining face.

The natural caving method was adopted for the goaf roof, and the advanced pre-splitting roof cutting method was adopted for the reserved roadway side. The ventilation mode was two inlets and one return.

There is no false roof in the coal seam of the 14,211 fully mechanized mining face. The direct roof of the coal seam is gray siltstone with a thickness of 3.1 m. The key layer of the basic roof is the interbedding of medium sandstone and medium-grained sandstone, with a thickness of 42.06 m. The average saturated compressive strength of the roof rock stratum is 23.7 Mpa, belonging to the unstable~relatively stable type (I~II). The mechanical parameters of the coal and rock mass are shown in Table 2, and the lithologic column within the upper and lower range of the coal seam is shown in Figure 4. In recent years, the model operation based on batch data verification has become popular. Some studies have used it to estimate the shear strength of rock joints, evaluate the stability of underground entrance excavation, and predict the mine safety evaluation system under the encouraged pillar strength, which has positive significance for mine safety mining and management [16–18].

**Table 2.** Mechanical parameters of the strata.

| Position | Rock Stratum | Thickness/ m | Bulk Modulus/ GPa | Shear Modulus/ GPa | Cohesion/ MPa | Tensile Strength/ MPa | Friction Angle/ (°) | Density/ (kg·m⁻³) |
|---|---|---|---|---|---|---|---|---|
| Main roof | Medium sandstone | 42.06 | 10.8 | 6.5 | 3.90 | 5.8 | 37 | 2755 |
| Direct Roof | Siltstone | 3.1 | 8.5 | 4.3 | 2.75 | 1.84 | 34 | 2460 |
| Coal seam | No. 4 | 3.88 | 0.99 | 0.35 | 1.0 | 0.5 | 28 | 1855 |
| Immediate bottom | Mudstone | 1.85 | 2.1 | 1.2 | 5.3 | 0.5–1.2 | 31 | 2150 |

| Histogram | Layer thickness(m) | Name | Lithology description |
|---|---|---|---|
| | 42.06 | Medium sandstone | Interbedded medium grained sandstone. The average saturated compressive strength is 23.7mpa. |
| | 3.1 | Siltstone | Dark gray siltstone, sandy texture, argillaceous cementation. |
| | 3.88 | No.4 Coal Seam | The coal is black, massive, semi bright type, weak asphalt luster, uneven fracture, endogenous fracture development, fractures are filled by calcite. |
| | 1.85 | mudstone | Sandy, agglomerate. |

**Figure 4.** Lithology histogram of roof and floor of 4# coal seam.

The fixed attribute of traditional coal mining methods (retaining protective coal pillars) leads to the waste of a large amount of coal resources during coal resource mining. This part of the lost resources may become permanent due to the characteristics of the goaf. The roof control of the 14,211 working face adopted shield support. In the original support process,

the roof adopted advanced support, and both the return air chute and transportation chute were 20 m ahead of the working face. In the mining process, the repeated movement of the advance support and the occupation of roadway space greatly affect the production efficiency of the mine, which also increases the construction technology and operation cost at the end of the roadway. The 110-roof cutting roadway protection method proposed by academician He Manchao is used to solve or reduce the impact of this problem on the Zhangjiamao coal mine. From the perspective of economic benefits and resource recovery rate, the 110 method greatly reduces the excavation cost of the fully mechanized mining face. The pillar-free mining technology adopted by the 110 method greatly improves the recovery rate of coal resources. Compared with the original mining method, the 110 method can reduce coal waste by 173,600 tons, which accounts for 4.9% of the coal reserves of the working face.

## 4. Principle, Advantages, and Process Design of the "110 Construction Method"

The so-called "110 construction method" refers to "1 working face, 1 roadway, and 0 coal pillars" [19–21]. The "110 construction method" is developed based on the "roof cutting short-arm beam theory" proposed by academician He Manchao; that is, directional roof cutting is carried out on one side of the goaf, and the roof is cut off by the mine pressure. The combination of high preload, constant resistance, and a large deformation anchor cable is used to control the stability of the roadway roof and ensure the stability and safety of the roadway surrounding rock on the goaf side. The local surrounding rock technology is adopted to realize the automatic cyclic roof cutting of the large-area roof. The pillar-free mining technology in this environment eliminates or reduces the occurrence of accidents and disasters. The mining technology of the bottomless coal pillar with roof directional presplitting and roadway automatic pressure relief has been successfully formed [22–24]. Some scholars have studied the external container composed of polyvinyl chloride (PVC) with large fracture strain and fiber-reinforced polymer (FRP) with a high strength to weight ratio. The addition of cement-based grouting material to the container can improve the axial deformation capacity, which can be used as a reference for similar working face support [25]. The field support form and principle diagram of the "110 construction method" are shown in Figure 5.

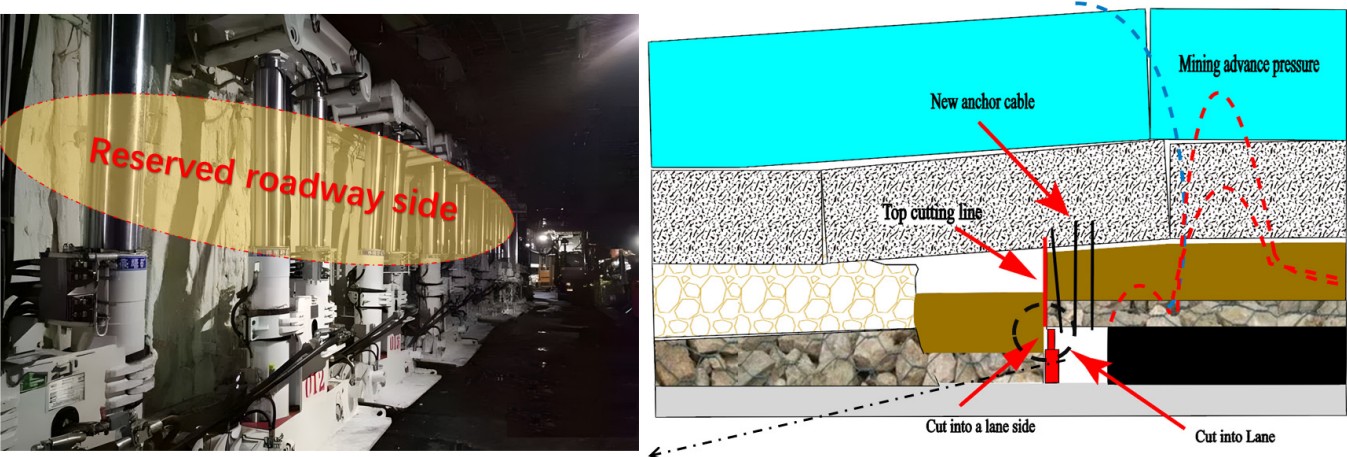

**Figure 5.** Schematic diagram and site picture of the "110 construction method".

Before the "110 construction method" is adopted, a ventilation system with two roadway air inlets and one roadway air return should be formed. The Zhangjiamao coal mine belongs to a coal seam with low gas and low spontaneous combustion. If two tunnels are used for air intake on the working face, it is better to adopt the Y-type ventilation in the next working face. The fresh air becomes dirty after flowing through the working face, and the dirty air flows out of the working face through the reserved roadway section. The

Y-type ventilation mode is shown in Figure 6. With the rapid development of the economy and the intensification of energy consumption, the mining depth of coal mines worldwide has increased yearly. Gas emission and accumulation have become a great obstacle to mine safety and efficiency; the use of Y-type ventilation is an effective way to ameliorate this problem [26,27]. Due to the stable safety protection and positive economic benefits of the "110 construction method", this technical measure has been successfully popularized and implemented in the mine.

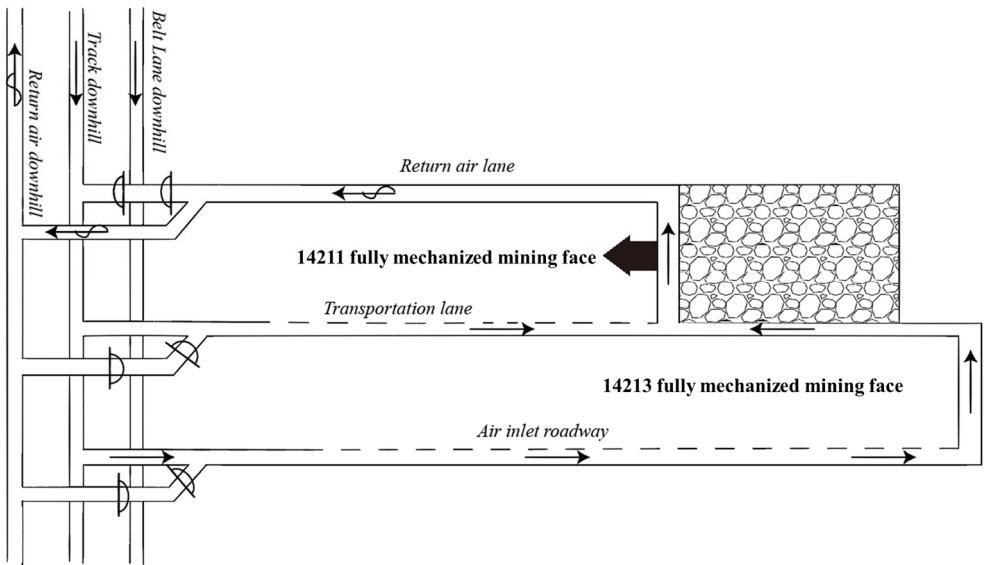

**Figure 6.** Schematic diagram of Y-type ventilation mode of the working face (top view).

## 5. Engineering Application and Effect Analysis

According to the design support parameters, the roadway support is strengthened. The transportation chute and the top plate of the chute shunting room were designed, and the "anchor cable + W steel belt support" was added, based on the anchor mesh beam support. All these measures prevent roof instability or roof fall of the reserved roadway section when the goaf roof is cut or periodic roof pressure occurs. Prestressed steel strands with a nominal diameter of φ 21.8 mm and lengths of 10 m and 9 m were used as the anchor cables; the roadway section support parameters are shown in Figure 7 and Table 3. Some studies have used discontinuous deformation to simulate surface settlement and achieved good results. The research verified that the joint inclination and excavation position were two important factors in the excavation design of underground space, because they affect the surface settlement and the stress concentration around the excavation area [28,29]; this research is helpful for us to carry out numerical simulation in the future.

**Table 3.** Support parameters of the working face transportation chute.

| Support Location | Support Mode | Support Material Parameters | Spacing × Row Spacing |
|---|---|---|---|
| Roof | Anchor cable, and W steel strip | Φ21.8 mm × 9000 mm. Φ21.8 mm × 10,000 mm (goaf side) | 1500 mm × 2400 mm 1500 mm × 1200 mm (goaf side) |
| Left side (Goaf side) | Bolt support Wire (reinforcing) mesh | Φ22.0 mm × 2200 mm. 50 mm × 50 mm (100 mm × 100 mm) | 900 mm × 1200 mm |
| Right side | Bolt support | Φ22.0 mm × 2200 mm. | 900 mm × 1200 mm |

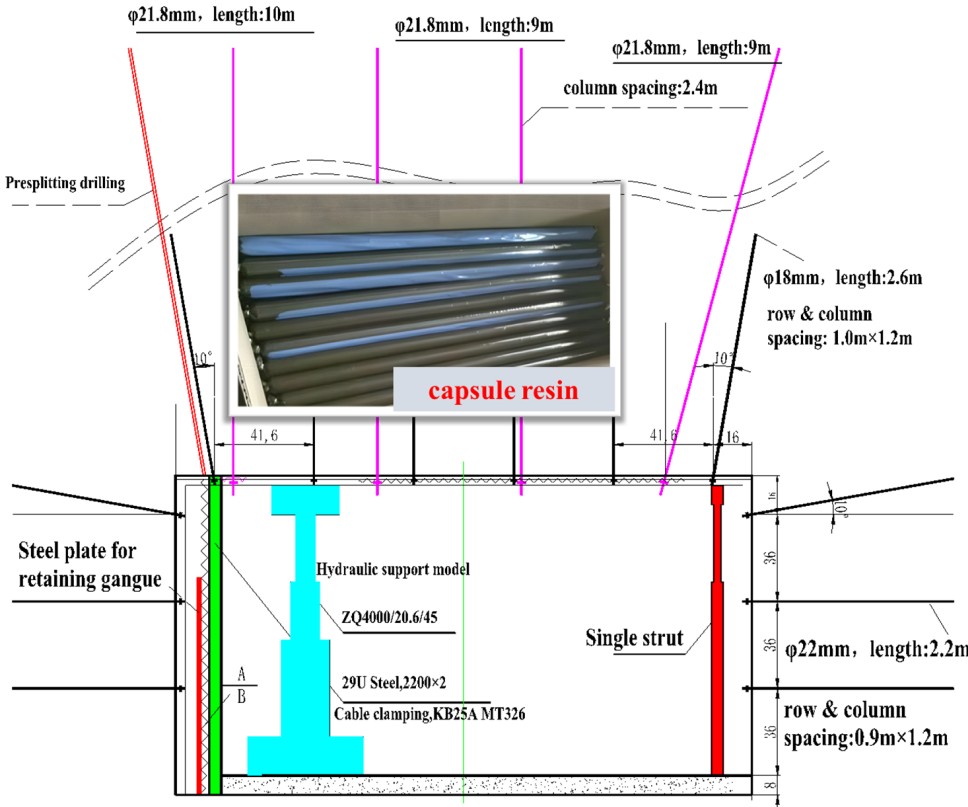

**Figure 7.** Schematic diagram of roadway support. A: φ 4 mm lead wire mesh, mesh size 50 mm × 50 mm. B: φ 6.5 mm reinforcing mesh, mesh size 100 mm × 100 mm. Cable row spacing and column spacing: 1500 mm × 1200 mm.

Each cable was anchored with one k2360 capsule resin and two z2360 capsule resins. The preload was no less than 280 kN, the ultimate tensile breaking force was not less than 583 kN, and the elongation was not less than 3.5%. The transportation chute was close to the primary mining side, the anchor cable spacing was 1200 mm, and the anchor point was 300 mm from the cutting top line and 600 mm from the main mining side. Three adjacent anchor cables were connected by a 2700 mm long W steel belt along the roadway. The row spacing between the other anchor cables was 1500 × 2400 mm, and the length of the supporting W steel strip was 3300 mm, which was arranged perpendicular to the roadway trend. The row spacing between anchor cables in the transport shunting chamber was 1250 × 2400 mm, the supporting W steel strip length was 2800 mm, and the anchor cable tray adopted an arched iron tray with a specific size of 300 × 300 × 16 (mm). In roadway excavation, the bolt support density was increased to ensure the surrounding rock stability of the roadway secondary mining side and the shunting room. The bolts used were glass fiber reinforced plastic bolts with a diameter of 22 mm and a length of 2200 mm. The support spacing and row spacing were 900 × 1200 mm; the top view of the roadway roof support is shown in Figure 8.

In this project, the bi-directional shaped charge blasting presplitting technology was adopted, and specific explosives were installed in the shaped charge device, which had the energy accumulation effect in two set directions. After the explosive was detonated, the surrounding rock pressure was uniformly compressed in the non-set direction but concentrated in the set direction. A microseismic (MS) system was used to monitor vibration signals and collect and analyze vibration signals in the process of drilling string construction. The purpose of "one hole for multiple purposes" can be achieved in drilling depth monitoring, bearing pressure distribution, and pressure relief effect evaluation, which can evaluate the mine pressure more accurately and effectively [30,31]. The GS-GMDH model proposed by some researchers has verified its ability to predict the ground vibra-

tion caused by blasting, which has guiding significance for the feasibility of predicting a blasting operation in advance [32]. According to the characteristic that rock is easily damaged by tension, the tensile fracture forming of rock mass in the set direction was realized. The calculated depth of the pre-crack hole was 8.25 m, the designed depth of the crack hole was 8.5 m, and the hole diameter was 50 mm. The slit hole was arranged 0.3 m away from the main mining side of the roadway, and the included angle between the top slit hole and the plumb line direction was 10° (towards the goaf). As the roof of the working face was mainly siltstone and fine-grained sandstone, the slit hole spacing of the roof was designed to be 600 mm, with one pre-split hole in each row. The pre-crack of advance blasting was ~70 m away from the working face. The W steel strip made the roof connection structure an interrelated whole, which played a better bearing role [33]. As an effective blasting method, pressure relief blasting is widely used. Some scholars use the mixed model of information entropy and unascertained measure with different membership functions to evaluate explosive areas. The results showed that the combined model can eliminate the interference of subjective factors and ensure the reliability of the evaluation results [34].

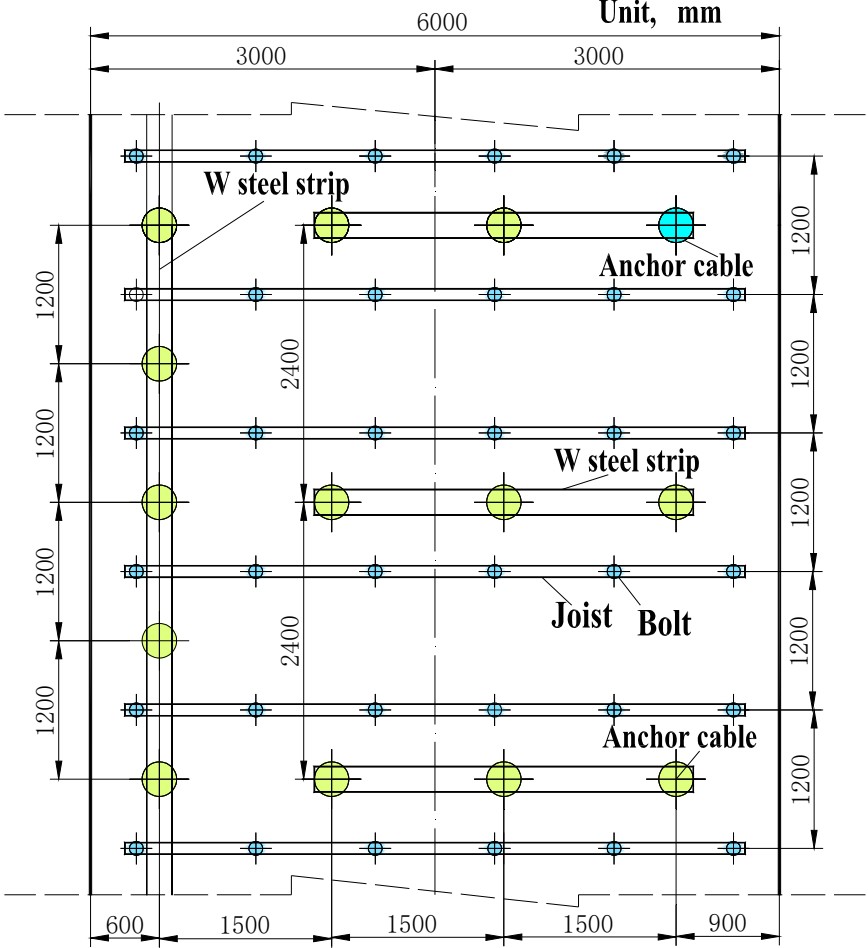

**Figure 8.** Top support plan of 14,211 and 14,212 transport chute (unit, mm).

The ground above the working face is an area without personnel activities. After implementing the design scheme, the surface settlement was small, and the impact of mining activities on the ground was not obvious. The relative position between the development trend of ground fissures and the working face is shown in Figure 9.

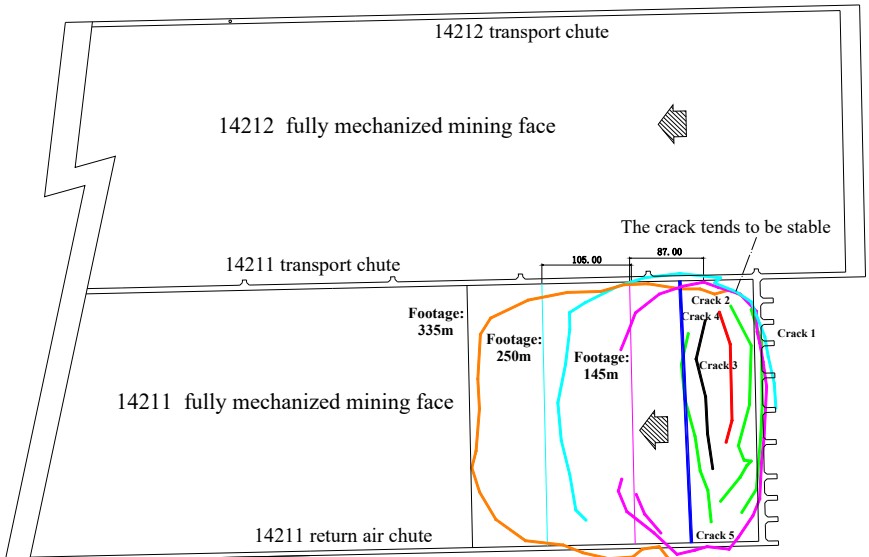

**Figure 9.** Working face layout and ground crack position.

The fracture distribution area photographed and depicted by the UAV was consistent with the roof activity law during the advancement of the underground working face. The initial collapse of the roof led to large surface subsidence at the cut hole position. When the working face entered the stage of cyclic top caving, the surface cracks gradually decreased. The farther away from the cut hole, the smaller the cracks became; the later roof collapse had a very slight impact on the surface. This is consistent with the gradual reduction and thinning of cracks photographed and drawn by UAV in Figure 9. Figure 10 shows the impact of the collapse and extension of the goaf to the ground during mining activity. The surface subsidence of the cut hole part was apparent, but the area was limited. The fracture area was only limited to the cut hole position. As the working face entered the normal cycle, the surface subsidence was weakened, and the surface fissures tended to be stable. The slight fissures and subsidence of the ground did not threaten the surface vegetation and ecology.

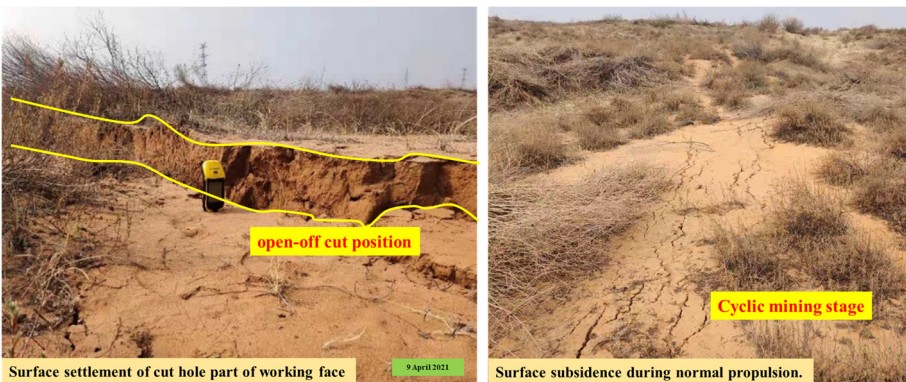

**Figure 10.** Comparison of surface subsidence between cut hole position and normal mining.

## 6. Conclusions

By implementing the 110-construction method, the coal recovery rate was improved, and the waste of resources was reduced. Field engineering verified the feasibility of the design scheme; and the main conclusions are as follows:

(1) The distribution of global coal resources is uneven, and the contradiction between social development and the intensified energy consumption is irreconcilable. In the

process of resource development, it is imperative to improve the resource recovery rate and reduce the waste of coal resources.

(2) The successful application of the 110-construction method in the Zhangjiamao coal mine avoided the waste of 173,600 tons of coal, with a value of RMB 152.814 million ($23.33 million), which provides a direct and beneficial experience for other coal mines.

(3) During the implementation of the 110-construction method, the maximum subsidence of the roof was 8.39 cm, and the average subsidence was ~4.5 cm; the roof was complete and reliable. There were no major safety accidents or casualties in the mining process.

(4) After the working face entered the normal mining process, the surface settlement was significantly reduced. Only small cracks extended to the surface, which did not threaten the ground vegetation and organisms. On the premise of safe production, the technology saved excavation quantities, reduced coal loss rate, improved production efficiency, and protected the surface ecology, so as to form a positive cycle of the technology.

**Author Contributions:** Z.L. and J.Z. designed and wrote the paper; X.S. supervised the paper writing; H.C., Y.Z. (Yanyang Zhang) and Y.Z. (Yanjun Zhang) collected and collated materials and field data collection. All authors have read and agreed to the published version of the manuscript.

**Funding:** This research was funded by the "Accurate delay rock breaking mechanism and key technology innovation team", grant No. 2020D14043.

**Institutional Review Board Statement:** Not applicable.

**Informed Consent Statement:** Not applicable.

**Data Availability Statement:** The data used to support the findings of this study are available from the corresponding author upon request.

**Conflicts of Interest:** We declare that we do not have any commercial or associative interest that represents a conflict of interest in connection with the work submitted.

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
