# Peer review of "A Safe and Efficient Mining Method with Reasonable Stress Release and Surface Ecological Protection"

_sustainability, doi:10.3390/su14095348_

Round 1
Reviewer 1 Report
This paper lists a new coal mining technology “110 construction method” and gives an application case. The description of engineering cases is relatively complete. However, the numerical simulation in this paper is too rough. Simulation parameters, boundary conditions and support parameters are not given.
Reviewer 2 Report
The paper seems to be a presentation of results obtained by the use of so called "110 construction method", which in esence refeers to the replacement of pillars between two adiacent longwall faces with a combined support using hydraulic supports and anchors. This technical solution emerging from a pattent https://patentscope.wipo.int/search/en/detail.jsf?docId=WO2016206617 , not cited in the paper's bibliography.
It is mainly a case study, and the main conclusions lead to the avoidance of waste coal (from pillars) and limited amount of subsidence and damaging of surface.
The introduction is to large, by my opinion, being related to the extent of coal as energy resource. The FEm model and the experimental (field ) results seems to be in accordance with the conclusions claimed. I think, the paper after the rewritting, with a clear introduction in the spirit of the research, avoiding generalities, could be published, because it is clearly an attempt to improve the knowledge on coal seams mining at reduced depth with reduced losses and reduced environmental impact in terms of surface damage.
Reviewer 3 Report
The draft of “Safe and efficient mining method of stress release based on surface ecological protection” provides an interesting discussion on the mining design. However, the following unclear parts may help the authors improve their draft.
- Title, “Safe and efficient mining method of stress release based on surface ecological protection”, the English seems strange because the surface ecological protection is the results of the mining method but not the theory of the mining method. Therefore, it is hard to image the use of “based on surface ecological protection”.
- Line 39, “200 billion yuan”, should the “yuan” be change to “RMB” as a formal money unit?
- Line 45, “Kang et al.”, check the format of the citation.
- In Fig. 3, the words are too small.
- Line 115, “The working face is a double lane”, what is the depth of the working face?
- Line 123, “…mining areas are shown in Figure 6.”, check the sequence of the figure. Why Fig. 6 shows up before Fig. 4 in the text?
- Line 149, “The 110-roof cutting roadway protection method proposed by academician He Manchao is used to solve or reduce the impact of this problem on Zhangjiamao coal mine.”, references regarding to 110-roof cutting roadway protection method must be added.
- In Fig. 6, the authors must clarify the figure is a side view or a top view?
- Line 194, “established by FLAC3D software”, which version is used? Add it to the text.
- Line 195, “The model size is finally determined as 20m × 25m × 0.5m (long × high × Width)”, since the width is 0.5 m, the authors must add additional statements to explain why the 3D simulations are required but not 2D.
- 8, what are the unit for the values on the color bar? In addition, why a light blue color is on the top and right of the excavated hole but the pink in the left side wall?
- What do shear-n, shear-p, tension-p, and tension-n mean in Fig. 9?
- Line 228, “The separation value of roof layer on site is 5-10cm, which is basically consistent with the numerical simulation results as shown in figure 11”, the authors must add additional statements about how to get appropriate parameters in the numerical simulations to get good results.
- 10, what are the unit for the values on the color bar?
- Line 298, “the roof activity law”, add additional statements to clarify the law.
- Line 316, “with a value of 152.814 million yuan”, should the yuan be change to RMB? In addition, the authors must also provide the value with US Dollars for the people who are not familiar with Chinese RMB.
- The following references used discrete element method to simulate the rock behavior during tunnel excavation. The authors must cite them and explain why this study can assume the rocks as continuous media but not jointed?
- Verifying discontinuous deformation analysis simulations of the jointed rock mass behavior of shallow twin mountain tunnels, International Journal of Rock Mechanics and Mining Sciencesthis link is disabled, 2020, 130, 104322
- Simulation of the inclined jointed rock mass behaviors in a mountain tunnel excavation using DDA, Computers and Geotechnicsthis link is disabled, 2020, 117, 103249
- Find a native speaker to sharpen the English of the draft.
Round 2
Reviewer 1 Report
In my opinion, the author can remove the part of numerical simulation in this paper, which is still not very rigorous. The description of the numerical method is still vague. In numerical simulation, the calculation model is very important, so please provide the mathematical model for calculation. Moreover, the authors should specify the numerical schemes (central differences? Upwind, QUICK? Minimod?) and the related accuracy (first order? Second order? Etc.). In the numerical simulation part, how to set the boundary conditions is not detailed enough. For example, how to set the boundary conditions between different partitions? How does the stress change near the interface? Is it a sudden change or a continuous change? Moreover, this model is actually a 2D model, how to consider the influence of working face advance on roadway stability.
Author Response
According to the reviewer's opinions, after discussion, we decided to delete the part of numerical simulation in the article and revise the description of numerical simulation in the full text. We also comprehensively checked and revised the article again. We will carefully consider the questions raised by the reviewer to have a deeper understanding and more systematic study of numerical simulation in the follow-up work. Thanks again for the detailed comments and guidance.Reviewer 2 Report
Now is ok.
Author Response
Thanks again for your detailed comments and guidance.
Reviewer 3 Report
Most of my questions were answered by the authors. They did a good job. However, before the acceptance of the paper, the authors should have the minor modification.
1. Add the unit to the side bar of Fig. 8 and 10. In addition, the unit should also be added to Fig. 13. With adding the unit, the readers can easily confirm the unit of each figure.
Author Response
We revised the article according to the comments. Due to the adjustment of the article content, we deleted figure 8 and figure 9 (figure number of the previous version). We added the unit (mm) at the top of figure 8 (figure 13 in the previous version). There is no adjustment in the references of the paper.
Thanks again for your detailed comments and guidance.
Round 3
Reviewer 1 Report
this paper can be accepted